# QUICK: Quality and Usability Investigation and Control Kit for Mass Spectrometric Data from Detection of Persistent Organic Pollutants

**DOI:** 10.3390/ijerph16214203

**Published:** 2019-10-30

**Authors:** Wenjing Guo, Jeffrey Archer, Morgan Moore, Jeffrey Bruce, Michelle McLain, Sina Shojaee, Wen Zou, Linda A. Benjamin, Anthony Adeuya, Russell Fairchild, Huixiao Hong

**Affiliations:** 1U.S. Food & Drug Administration, National Center for Toxicological Research, 3900 NCTR Road, Jefferson, AR 72079, USA; Wenjing.guo@fda.hhs.gov (W.G.); Wen.Zou@fda.hhs.gov (W.Z.); 2U.S. Food & Drug Administration, Arkansas Laboratory, 3900 NCTR Road, Jefferson, AR 72079, USA; Jeffrey.Archer@fda.hhs.gov (J.A.); Morgan.Moore@fda.hhs.gov (M.M.); Jeffrey.Bruce@fda.hhs.gov (J.B.); Michelle.McLain@fda.hhs.gov (M.M.); Sina.Shojaee@fda.hhs.gov (S.S.); Russell.Fairchild@fda.hhs.gov (R.F.); 3U.S. Food & Drug Administration, Center for Veterinary Medicine, 7500 Standish Place, Rockville, MD 20855, USA; Linda.Benjamin@fda.hhs.gov; 4U.S. Food & Drug Administration, Center for Food Safety and Applied Nutrition, 5001 Campus Dr, College Park, MD 20740, USA; Anthony.Adeuya@fda.hhs.gov

**Keywords:** persistent organic pollutants, quality control, data usability, gas chromatography–mass spectrometry (GC–MS)

## Abstract

Persistent organic pollutants (POPs) cause a significant public and environmental health concern due to their toxicity, long-range transportability, persistence, and bioaccumulation. The US Food and Drug Administration (FDA) has a program to monitor POPs in human and animal foods at ultra-trace levels, using gas chromatography coupled with mass spectrometry (GC–MS). Stringent quality control procedures are practiced within this program, ensuring the reliability and accuracy of these POP results. Due to the complexity of this program’s quality control (QC), the decision-making process for data usability was very time-consuming, upward of three analyst hours for a batch of six extracts. We significantly reduced this time by developing a software kit, written in Python, to evaluate instrument and sample QC, along with data usability. A diverse set of 45 samples were tested using our software, QUICK (Quality and Usability Investigation and Control Kit), that resulted in equivalent results provided by a human reviewer. The software improved the efficiency of the analytical process by reducing the need for user intervention, while simultaneously recognizing a 95% decrease in data reduction time, from 3 hours to 10 minutes.

## 1. Introduction 

Environmental contaminants that reach our food supply, such as persistent organic pollutants (POPs) [1,2,3,4,5,6], are associated with many health concerns, such as hormone disruption, cancer, cardiovascular disease, obesity, reproductive and neurological ailments, learning disabilities, and diabetes [1,7,8]. The Stockholm Convention [9,10,11], an international regulation that requires participating countries to eliminate or reduce the release of key POPs in the environment, was introduced to help mitigate these health risks. Methods for the analytical determination of many of these compounds require comprehensive quality control (QC), which results in increased turnaround times [12,13,14,15].

There have been efforts focused on increased efficiencies (reduced turnaround time and cost) over the years. Most of these, however, have involved investigating alternate instrumentation [16,17,18,19,20], extraction techniques [9,10,21,22,23,24,25], and extract cleanup methodologies [26,27]. More recently, there have been developments in data evaluation, such as evaluating patterns of large data sets [28]. Quality criteria within the European Union (EU) regulation for polychlorinated dibenzo-p-dioxins, polychlorinated dibenzo furans (D/Fs), and polychlorinated biphenyls (PCBs) uses the ratio between the lower bound (LB) and upper bound (UB) toxic equivalency (TEQ), to assist in determining data acceptability [15].

Python scripting language has been widely used in the data analysis field because of its abundant libraries and ease of use. Recently, some researchers have developed python-based tools to assist in analyzing mass spectrometric data [29,30,31]. These implementations have proven to greatly increase the efficiency by automating time-consuming data-processing actions. Therefore, to improve the efficiency in detection of POPs using GC–MS systems, we developed a python-based customized software kit, QUICK (Quality and Usability Investigation and Control Kit), to investigate the quality and usability of data yielded from QC and target samples.

Multiple criteria are used for the determination of data usability within the FDA’s POPs program. The determination of contaminants in food requires stringent quality assurance (QA) and quality control (QC) procedures to help ensure data reliability [32]. Direct isotope dilution methods [12,13,14,15] are used for the targeted analytes, thus allowing an internal standard recovery calculation. Additionally, to reasonably accept results, detection limits, absolute responses, chromatography, lack of interferences, and associated QC concerns, such as no elevated levels found in the method blanks and spiked statistics, are evaluated. We have created a series of algorithms to convert our data coming from multiple vendors into a format to compare the above criteria. The focus of our work includes the use of LB/UB comparison to calculate a relative number, identified as a usability factor (UF), while adding data quality objective (DQO) to assess the data quality of the analytical results.

## 2. Method

### 2.1. Study Design and Overall Workflow of QUICK 

This study aims to automate the evaluation process of data quality and usability in POPs detection by GC–MS so laboratory scientists (referred as human experts in this paper) do not need to manually interpret data quality metrics for decision making. QUICK is expected to generate all metrics for investigating quality and usability of GC–MS experimental data and to suggest decisions for human experts to consider. 

The process of experimentally generating data and assessing quality and usability of data is depicted in Figure 1. Before a batch of target samples are subjected to GC–MS, some QC samples are used to test and ensure instrument optimization and that extraction methods are adequate. The data generated from QC and target samples are manually investigated by human experts to determine GC–MS instrument stability and to evaluate the quality and data usability of target sample analysis. This process is very time-consuming. Therefore, QUICK was designed to automate the investigation of data quality and usability for QC of POPs detection. The overall workflow of QUICK is illustrated in the gray box of Figure 1. 

### 2.2. QC Samples

Two types of samples are analyzed for detection of POPs: (1) QC samples to determine data quality and usability and (2) target samples for POPs determinations. There are four types of QC samples. The first, Daily Standard (termed as Standard hereafter), is a standard solution ready to inject into the instrument and used for instrument preparedness. The Standard contains known amounts of each congener. The congener concentrations are equal to the midlevel standard of the 5-point calibration curve. This Standard solution is analyzed prior to a sample batch (other QC samples and target samples), every 24-hours, and used to verify instrument suitability. Many quality criteria are verified from this injection, including response factors, absolute retention times, relative retention times, and absolute responses to determine instrument reliability [33,34]. The second QC sample is the method blank (termed as Blank hereafter). Blank, a 1 g aliquot of corn oil, is used to verify that levels detected are below reporting limits. This Blank oil is treated identically to target samples extracted in the same batch, including exposure to all glassware, equipment, solvents, reagents, and internal standards. Blank is used to demonstrate that the system (laboratory, reagents, glassware, etc.) is free of contaminants and ensure that the extraction process is under control. Method Spike (termed as Spike hereafter) is the third type of QC samples. Spike is similar to Blank but contains a known amount of all method analytes and is used for ongoing accuracy calculations. The final QC sample is Duplicate, which is an extracted and analyzed target sample replicate and used to determine method precision.

Standard is analyzed first to determine instrument suitability. If Standard passes QC criteria, extraction batch QC and target samples are analyzed. Blank is extracted and analyzed with every batch, while Spike and Duplicate are extracted and analyzed monthly, at minimum. If data from Standard fails to meet the quality criteria, instrument maintenance may be needed and Standard reanalyzed.

### 2.3. Reference Data

Human experts evaluate current QC data against 100 historically obtained reference QC data points to verify that QC samples are compliant. All QC results should be statistically similar, within ± 3SD of the historical reference data, except blank results, which should be within ± 2SD of the blank reference data. QUICK was designed to rapidly evaluate these criteria. Some reference data (Blank and Spike reference for PCBs/PBDEs) were collected by human experts and manually compiled. This type of reference data requires extra maintenance and upkeep. Other reference data (Standard and Spike reference for dioxin/furan), were compiled with the help of preprocessing packages (TargetLynx and ChromaTOF). QUICK follows the same pattern used by the human experts in their manual evaluation process. 

### 2.4. Standard Data Quality

Before evaluating data quality, including Standard, Blank, Spike, Target, and Duplicate, all results must be in the format requested by QUICK; otherwise, QUICK is unable to interpret these data. This will result in output warnings, causing QUICK to halt the data quality evaluation. When the format is correct, QUICK assesses the quality of MS data. 

Prior to analyzing any extracted QC or target sample, instrument calibration is performed to demonstrate instrument stability. Standard is analyzed and preprocessed using the vendor-provided software package (TargetLynx or ChromaTOF). The concentration of each congener is compared to the concentrations of the same congener of the samples in Standard Reference to verify quality. Human experts examine data quality by using a control charting method from TargetLynx for dioxin/furan congeners. The control chart graphically displays the concentration for each of the dioxin/furan congeners in Standard Data, with the control limits, mean ± 3 standard deviations (SD) in Standard Reference. The human experts manually check the control chart to determine instrumental acceptability. However, a similar function does not exist with ChromaTOF, causing human experts to manually check the concentration output for both Standard Data and Standard Reference, to determine if the instrument is performing optimally. Only when Standard Data demonstrate that all congener concentrations are in the allowed range defined by Standard Reference, human experts continue to work on QC and target samples. Otherwise, intervention is required and then Standard is reinjected. QUICK was designed to automate the investigation of Standard Data quality. It reports the detailed quality metrics and recommends QC action (pass or fail) to human experts (Figure 1). QUICK also compares the Standard Data with the Standard Reference to determine if the Standard Data should be included in the Standard Reference. 

### 2.5. Blank Data Quality

A Blank is extracted with every sample batch, consisting of Target Data and potentially Spike and Duplicate Data. Blank Data is compared to the Blank Reference of previously tested oil that demonstrates no detected levels greater than accepted blank limits. Blank Data are used for verifying that no contamination occurred during the sample preparation process. In the laboratory, Blank sample is analyzed to generate the corresponding Blank Data. Since Blank contains little to no analytes of interest, the amounts of analytes identified should be below the blank limit, which is the mean + 2SD determined by Blank Reference. The criterion used for Blank Data quality investigation is that the concentration of a congener should be lower than the corresponding blank limit. QUICK was designed to compare the concentration of each analyte with the blank limit. In the same way, for Standard Data quality, QUICK flags the congener in this step if the detected amount is greater than its blank limit.

### 2.6. Spike Data Quality

Spike is tested to measure the accuracy of the experimental procedure and is extracted and analyzed on a monthly basis, at minimum. The evaluation of Spike Data quality is very similar to investigating Standard Data quality. Human experts use control charts to check the quality of dioxin/furan data and manually compare Spike Data with Spike Reference for PCB/BDE. QUICK was designed to automate the quality assessment of Spike Data in a similar way to Blank Data quality. It compares Spike Data with the data in Spike Reference to determine whether the concentrations of analytes were within the range of mean ± 3SD of Spike Reference. If the amount of an analyte is outside the range, QUICK flags the congener in its report to users.

### 2.7. Target Data Usability without Duplicate Data

If unexpected results (amounts detected or limits of detection (LOD)) are discovered for congeners, Target Data usability is problematic, and human experts may need to re-extract and reanalyze the target sample. 

Before evaluating data usability, Target Data are preprocessed using vendor software. If the ion ratio is deemed acceptable via preprocessing, no change is made by QUICK. If the ion ratio fails, however, the concentration is set to zero, thus elevating the LOD to an Estimated Maximum Possible Concentration. At any point, a concentration that is greater than a Blank Reference is changed to zero by QUICK, the corresponding LOD is elevated to the amount found. If the concentration of a congener in Blank Data is larger than the blank limit from Blank Reference and the concentration in Target Data is smaller than five times of the concentration in Blank Data, this congener is labeled as non-detect and the LOD is elevated to the amount found [25]. If the concentration of the congener in Target Data is smaller than the blank limit from Blank Reference, this congener is labeled as non-detect and the concentration is set to zero by QUICK. For dioxin and furan, a congener is labeled as non-detect and its concentration is set to zero if its relative retention time is not within the EPA limits [12] or if the signal-to-noise ratio is less than five. Otherwise, no change is made to the concentration. Concentrations are set to a value of zero rather than non-detect, to allow the calculation of lower bound and upper bound concentrations/toxic equivalency (TEQ). A sum of all concentrations/ TEQs are used to calculate the lower bound concentration/TEQ of a sample, whereas, if the concentration/TEQ is zero, the LOD is used in the place of concentration/TEQ to calculate the upper bound concentration/TEQ. Therefore, this gives a best-case scenario (lower bound) assuming zero found were truly zero found, while the upper bound is a worst-case scenario that suggests the LOD was detected and included into the calculation. If the upper bound (worst-case) is less than the level of interest, C_UB_, then the data are useable, as shown in the first step of Figure 2. The C_UB_ is a variable concentration that is dependent upon the assignment and data quality objectives.

The preprocessed concentrations and LODs are used to calculate data quality metrics, such as the lower bound and upper bound, usability factor, and congener contribution. The usability factor (UF) is calculated by Equation (1): (1)$$\mathrm{Usability}\;\mathrm{Factor}=\;\frac{\sum_{i}\mathrm{Upper}\;\mathrm{Bound}-\sum_{i}\mathrm{Lower}\;\mathrm{Bound}}{2\sum_{i}\mathrm{Blank}\;\mathrm{Limit}\;+\;\sum_{i}\mathrm{Lower}\;\mathrm{Bound}}$$

The TEQ is used to express the toxicity of a sample, and it is calculated by the product of the concentration and toxic equivalency factor (TEF) [35] values of each congener. The PBDEs and marker PCBs do not have related TEF values and are thus calculated with upper bound and lower bound concentrations. Blank limit is calculated from Blank Reference data. It is either TEQ or concentration, depending on the congener type. 

The congener contribution (CC) is calculated based on the congeners that have concentrations of zero and LOD > blank limit, using Equation (2): (2)$$\mathrm{Congener}\;\mathrm{Contribution}\;=\;\frac{\sum_{i}\mathrm{LOD}}{\sum_{i}\mathrm{Upper}\;\mathrm{Bound}}$$

Using the same strategy as human experts, QUICK determines the usability of Target Data as depicted by the flowchart shown in Figure 2. The objectives for the usability factor (C_UF_) and congener contribution (C_CC_) are variables chosen by experts depending upon the data quality objective of the assignment and target samples being analyzed. If the usability factor shown in Figure 2 is less than 0.5, that means that congeners were detected with minimal elevated detection limits. If the usability factor is greater than 0.5, then some elevated detection limits were reported for the sample. Again, depending upon the assignment, C_UF_ values between 0.5 and 0.65 are commonly used. The C_CC_ identifies the percentage of problem analytes if the usability factor is greater than 0.5. A C_CC_ value of 0.1 is commonly used, indicating less than 10% of the congener contribution can come from elevated detection limits. Table 1 shows examples of objective values for various matrices and analyte types (for the matrix not listed in Table 1, the objective of sample upper bound is set to zero for all congener types). As illustrated in Figure 2, if the sample upper bound is lower than its objective, C_UB_, target data are usable. If not, the usability factor of Target Data is compared with the corresponding objective. Target Data are usable when the usability factor is smaller than the objective; otherwise, QUICK checks the congener contribution. If the congener contribution is smaller than the objective, Target Data are usable; otherwise, Target Data are not usable. In other words, the Target Data must fail all three tests to be considered unacceptable.

### 2.8. Target Data Usability with Duplicate Data

Duplicate Data are generated from a replicate aliquot of target sample, monthly at minimum. It is performed to measure the precision of the test procedures and usability of Target Data. QUICK calculates two relative percent difference (RPD) values to evaluate the quality of Duplicate Data and usability of Target Data. 

QUICK calculates the RPD based on the concentration for PBDE and marker PCB, using Equation (3): (3)$$RPD_{i}^{Con}\;=\;\frac{\left(C_{i}^{DUP}-\;C_{i}^{TAR}\right)}{\left(C_{i}^{DUP}+\;C_{i}^{TAR}\right)/2}\;\times 100\%$$
where $RPD_{i}^{Con}$ is the RPD based on concentration for congener *i*; $C_{i}^{DUP}$ is the concentration (pg/g) of congener *i* in Duplicate Data; and $C_{i}^{TAR}$ is the concentration (pg/g) of congener *i* (pg/g) in Target Data. 

For the remaining congeners, QUICK computes RPD based on TEQ for each congener, using Equation (4): (4)$$RPD_{i}^{TEQ}\;=\frac{\left({\mathrm{TEQ}}_{i}^{DUP}-\;{\mathrm{TEQ}}_{i}^{TAR}\right)}{\left({\mathrm{TEQ}}_{i}^{DUP}+\;{\mathrm{TEQ}}_{i}^{TAR}\right)/2}\;\times 100\%$$
where $RPD_{i}^{TEQ}$ is the RPD based on TEQ for congener *i*; $TEQ_{i}^{DUP}$ is the TEQ of congener *i* in Duplicate Data; and $TEQ_{i}^{TAR}$ is the TEQ of congener *i* in Target Data. 

If more than five congeners (a parameter set by human experts) have RPD values larger than 25%, the Target Data and Duplicate Data are considered low quality. QUICK reports all calculated RPD values, marks the congeners with an RPD larger than 25%, and makes data-quality assessment conclusions.

### 2.9. Implementation

QUICK was developed in Python 3.6 (http://python.org) (Python Software Foundation, Beaverton, OR, USA). Some extra modules, such as tabular and PyPDF2, were added to the anaconda environment to help input PDF files. The graphical user interface was built with the Qt5 framework bound via PyQt5. The source code and Windows executable files are available upon request.

## 3. Results

After implementation, QUICK was integrated into the routine FDA POPs laboratory and tested for several months. To evaluate the performance, human experts repeated the QA/QC process, and the results were compared with those from QUICK. The testing results revealed that QUICK produced very similar QA/QC results to those obtained from human experts. Furthermore, QUICK reports more detailed QA/QC metrics and suggestions on QC action. With QUICK, the data evaluation time for POPs detection using GC–MS drops dramatically. Testing results from QUICK are reported and compared with the results obtained by human experts and are discussed below.

### 3.1. Standard Data Quality

During evaluation of Standard Data quality for the testing cases, QUICK yielded very similar QC metric values to those generated by the human experts. Here, the standard sample P180629-CS3A (tested on 29 June 2018 in FDA Dioxin lab) was taken as an example to show the performance of QUICK. The data quality metrics of the standard sample P180629-CS3A calculated by QUICK are shown in Figure 3, where the congener numbers on the *x* axis are given in the first column of Appendix A. The standardization scaling transformed concentrations of the 40 congeners depicted on the *y* axis are used for assessing quality of the Standard Data. The concentrations of all congeners are within the allowed range (mean ± 3SD), indicating the instrument worked properly. Human experts confirmed the Standard Data of sample P180629-CS3A are acceptable, as they found that all 40 congeners are within 3SD in the 40 control charts. As an example, the control chart for ^13^C-OCDD used by the human experts is shown in Appendix A. The concentration of ^13^C-OCDD is within 2–3 SD from both QUICK (the red bar in Figure 3) and the human experts (the rightmost point in Appendix A), indicating that QUICK replicated the QC result of the human experts. 

### 3.2. Blank Data Quality

Blank results play an important role in assuring that no contribution is made from the experimental method and assist in assessing the usability of Target Data. To show the performance of QUICK, the results of six Blank Data sets are presented here. QUICK sets concentrations to zero if the qualitative criteria for signal to noise ratio, ion ratio, relative retention time are not met (detailed in the method section). Upper bound concentrations of the 40 congeners detected in the 6 Blank samples from QUICK are compared with those obtained from the human experts in Figure 4. The correlation between the two results is very good with a correlation coefficient > 0.997. The relative difference in concentrations between the human expert and QUICK was calculated for each of the 40 congeners, using Equation (5):(5)$$\frac{X_{\mathrm{QUICK}}-\;X_{\mathrm{Human}\;\mathrm{Expert}}}{(X_{\mathrm{QUICK}}+\;X_{\mathrm{Human}\;\mathrm{Expert}})/2\;}$$
where X_QUICK_ and X_Human Expert_ represent the concentrations for congener X from QUICK and human experts, respectively. The relative difference values are <0.5% for all the congeners and samples, indicating that QUICK can reproduce the QA results from human experts. 

### 3.3. Spike Data Quality

In the assessment of Spike Data quality, QUICK compares the concentration of each congener with Spike Reference. Spike XS180905 is shown as an example to demonstrate quality assessment result for Spike Data using QUICK. The concentrations of the 40 congeners in standardization scale were plotted in Figure 5. The solid lines represent the region that meets quality requirement that is the mean ± 3SD, determined by Spike Reference. There are 13 congeners whose concentrations are out of acceptance criteria. QUICK identifies “failed” congeners with a pop-up window to warn that data quality of Spike is out of control. Using control charts, the human expert confirmed the Spike Data for sample XS180905 are not of good quality because the control charts for the same 13 congeners showed their concentrations are outside of acceptable limits. For example, control chart results for congener 1234789-HpCDF used by the human expert is presented in Appendix A. Comparing concentrations of this congener from QUICK (red bar in Figure 5) and human expert (the rightmost data point in Appendix A) indicates that QUICK yielded the similar quality metrics and the same quality assessment as the human expert, revealing the software kit is reliable for Spike Data quality assessment. 

### 3.4. Target Data Usability without Duplicate Data

Three quality metrics (upper bound, usability factor, and congener contribution) are calculated for Target Data and compared to the objectives of quality metrics in QUICK. The 69 Target Data sets from the most recently tested 26 samples were used to show Target Data usability assessment in QUICK. The quality metrics (sample upper bound, usability factor, and congener contribution) calculated by QUICK, using Equations (1) and (2) for the 69 Target Data sets are compared with their objectives. The results are plotted in Figure 6 as differences between quality metrics and their objectives for data upper bound (red bars), usability factor (blue bars), and congener contribution (green bars). The data usability decision flow shown in Figure 2 indicates that, when all three quality metric values of a Target Data set are larger than their objectives, the Target Data are considered problematic; otherwise, the Target Data are assessed as high quality. None of these 69 Target Data sets had all three quality metrics values larger than the corresponding objectives. QUICK reported them as high quality. The human experts assessed the same 69 Target data sets and concluded all data were acceptable, indicating QUICK is a reliable tool for the assessment of Target Data usability. 

### 3.5. Target Data Usability with Duplicate Data

Duplicate samples are tested monthly to evaluate precision. In evaluation of Target Data usability with Duplicate Data, relative percent difference (RPD) is used to measure consistency between Target Data and Duplicate Data provided from the same sample. The consistency is used as a quality metric of Target Data. QUICK uses the criterion of RPD less than 25% to determine the usability of Target Data. Sample S01 is used to demonstrate the performance of QUICK. Data usability for the Target Data and the Duplicate Data were evaluated separately, using the data quality decision flow in Figure 2. The result shown in Figure 7 indicates that both the Target Data and the Duplicate Data are usable regarding dioxin/furan, mono PCB, marker PCB, and PBDE data. However, the comparison between the Target Data and the Duplicate Data found they are not consistent in terms of RPD criteria. Figure 8 shows that 25 of the 42 congeners (including dioxins/furans, PCBs and PBDEs) for Sample S01 have unacceptable RPD values, greater than 25%. The elevated RPDs indicate that duplicate data need further investigation by human experts to resolve the difference. The RPD values of individual congeners calculated by the human experts, shown on the *x* axis in Figure 8, and the QUICK results, shown on *y* axis in Figure 8, demonstrate acceptable agreement between the two methods. 

## 4. Discussion

Although it is a very complex and time-consuming process, quality control in POPs detection at the laboratory level is critical in determining instrument performance and data usability. To improve efficiency, we developed QUICK to automate and execute the FDA’s quality control protocol. Results obtained from QUICK were consistent with results from expert analysts. The tests confirmed that QUICK gave equivalent results to the human reviews and provided reliable QA/QC results. Furthermore, QUICK reduces the data evaluation time from three human-expert hours to less than 10 minutes for a sample batch (typically six samples). QUICK greatly speeds up evaluation of the quality control procedure. In the development of POPs detection in labs, much attention is focused on improving the preparation and extraction methods and less attention is given to improving data evaluation. This software greatly improves the efficiency of reporting POPs in human and animal foods. QUICK allows analysts to work on more samples and provides reliable QA/QC results. The introduction of QUICK to the laboratory dramatically reduced the time for QA/QC of GC–MS data for POPs detection, improving FDA surveillance of POPs in human and animal food, thus, better protecting the public health. 

The data input for QUICK are produced from the vendor’s software system; thus, all QA/QC results from QUICK are based on the original, preprocessed data from the GC–MS systems. The QA/QC issues for generation of the original data are not within the scope of the current version of QUICK. Our results for data usability with Duplicate Data show that both Target Data and Duplicate Data would be in good quality if evaluated separately (Figure 7). However, comparisons of the two data sets show that they are not consistent for most of the congeners (Figure 8). Therefore, the Target Data, Duplicate Data, or both are problematic. The current version of QUICK cannot determine the cause of the problem, because the original data may have quality issues in generation that are not detectable. Current practice using Duplicate Data periodically not only increases the cost but also the time required for POPs detection. However, the Duplicate Data are an essential portion of thoroughly assessing the quality system, and our results warrant more frequent analysis of duplicate QC samples. The QA/QC generation of the original GC–MS data could help the downstream data quality assessment, thus deserving further investigation for the next version of QUICK. 

QUICK does not have limits on number of samples that can be processed in a batch. It can simultaneously assess data from as many samples as needed. This means that once GC–MS data are obtained and all QA/QC parameters are set, QUICK can evaluate the usability of all data at one time, in a single, run without user intervention. This feature is very useful since it reduces the possible errors in data entry and saves analysts time. 

Much of the QA/QC for POPs detection (as well as many other programs) are based upon samples, matrices, and instrumentation. For example, in the evaluation of data usability, the values of three quality metric objectives, upper bound objective, usability factor objective and congener contribution objective are dependent upon the matrix and assignment of the sample. With QUICK, users input the expert-determined parameters for the sample to conduct the QA/QC process. Therefore, the current version of QUICK is not a 100% automated process. Integration of machine-learning algorithms with huge amounts of data generated in the laboratories in the future may allow QUICK to automatically recognize the type of sample and matrix and determine QA/QC parameters, further improving the efficiency for POPs detection. 

## 5. Conclusions

To conclude, QUICK has improved the efficiency of the analytical process by reducing the need for user intervention while simultaneously recognizing a 95% decrease in data reduction time, from 3 hours to 10 minutes. There is a total of 75 dioxins, 135 furans, 209 PCBs, and 209 PBDEs. Only 7 dioxins, 10 furans, 18 PCBs, and 7 PBDEs are targeted for detection at the FDA POPs laboratory using GC–MS because of their toxicological significance. The current version of QUICK was developed specifically to facilitate detection of these toxicologically significant POPs; however, it can be modified to automate the QA/QC process of using GC–MS and liquid chromatography–MS for the detection of other environmental, food, and feed contaminants, like polyaromatic hydrocarbons and perfluoroalkyl substances. The complexity of environmental matrices also requires comprehensive QA/QC procedures to ensure the reliability of the data. QUICK can help automate this process with valid QA/QC criteria. In addition to human and animal food matrices, the incorporation of different POPs and QA/QC procedures will make QUICK a powerful tool for detection of POPs in the environment. 

## Figures and Tables

**Figure 1 ijerph-16-04203-f001:**
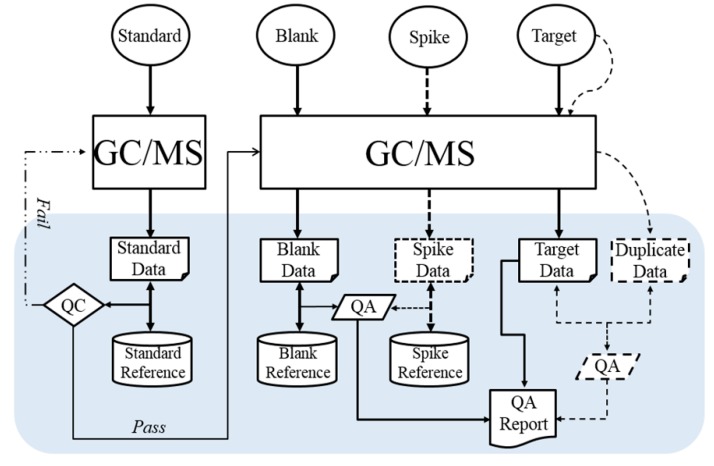
The process of experimentally generating data and assessing quality and usability of data. Dashed lines indicate optional procedures. Samples include both QC (Quality Control) samples (Standard, Blank, Spike, and Duplicate) and Target samples. Samples are injected into GC–MS (Gas Chromatography–Mass Spectrometry) to generate the corresponding data. The overall workflow of QUICK (Quality and Usability Investigation and Control Kit) is illustrated in the grey box. Standard Data are compared with Standard Reference to generate QC report. Blank, Spike, Duplicate, and Target data are used to report QA (Quality Assessment) results.

**Figure 2 ijerph-16-04203-f002:**
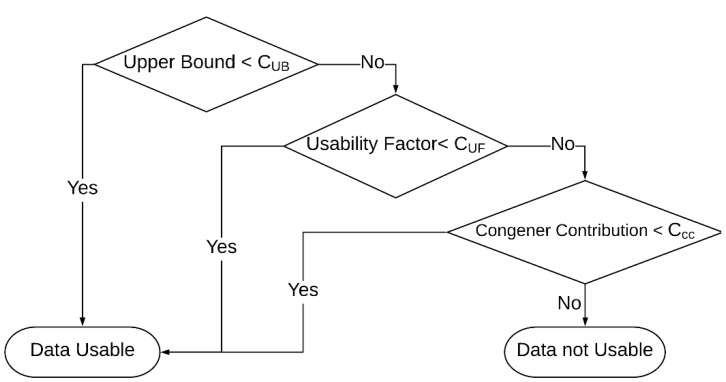
The flowchart for determining the usability of Target Data. $C_{UB}$, $C_{UF}$, and $\;C_{CC}$ are sample upper bound objective, usability factor objective, and congener contribution objective, respectively.

**Figure 3 ijerph-16-04203-f003:**
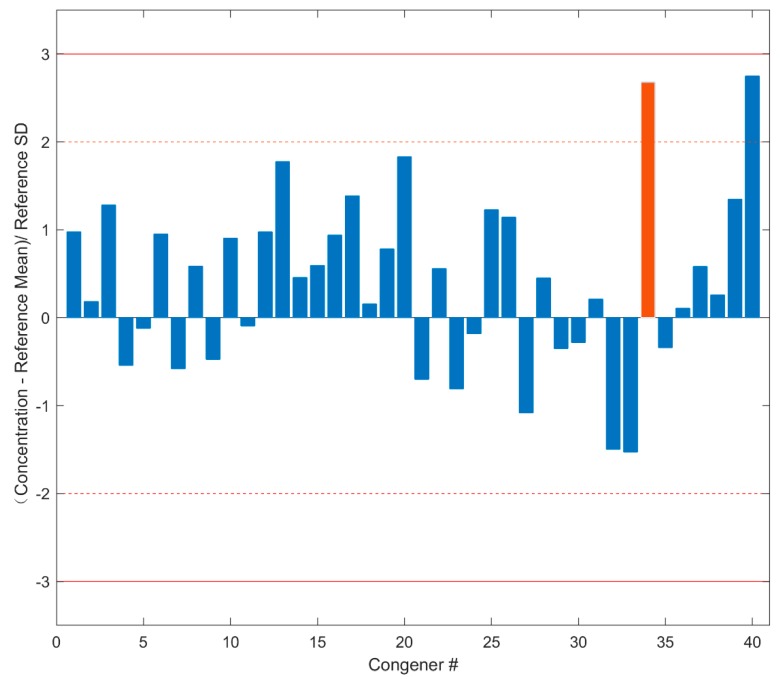
Concentrations of congeners in Standard Data. The differences in concentrations of the 40 congeners between Standard Data and corresponding mean concentrations in Standard Reference are transformed by standardization scaling to the standard deviations calculated from Standard Reference and are plotted on the *y* axis. The *x* axis indicates the congener numbers that are given in Appendix A. The dashed lines give the region within two standard deviations, and the solid lines depict the region within three standard deviations. The red bar represents the scaled concentration of congener ^13^C-OCDD.

**Figure 4 ijerph-16-04203-f004:**
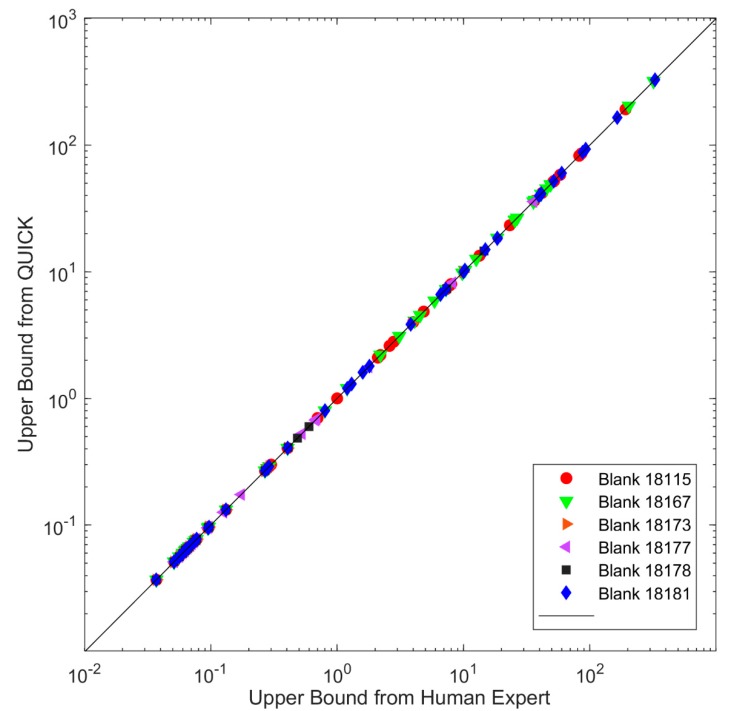
Comparison between the upper bound from QUICK (*y* axis) and upper bound from the human expert (*x* axis) for congeners of six Blank Data sets (solid red circles: data from Blank 18115; solid green down triangles: data from Blank 18167; solid orange right triangles: data from Blank 18173; solid purple left triangles: data from Blank 18177; solid black squares: data from Blank 18178, solid blue diamonds: data from Blank 18181). If a data point is on the diagonal line, the concentrations from QUICK and the human expert are the same for the congener.

**Figure 5 ijerph-16-04203-f005:**
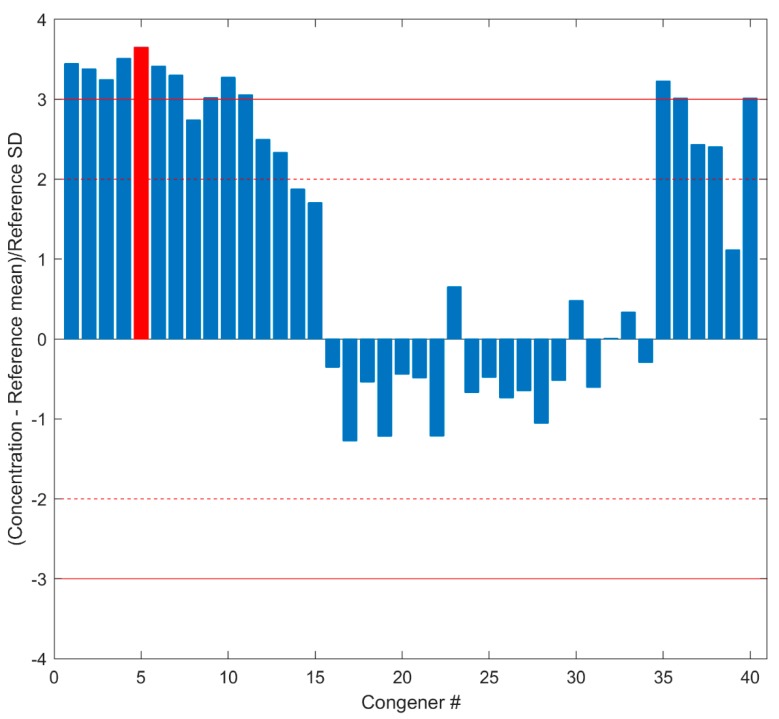
Concentrations of congeners in Spike Data. The differences in concentrations of the 40 congeners between Spike Data and corresponding mean concentrations in Spike Reference are transformed by standardization scaling to the standard deviations calculated from Spike Reference and are plotted at the *y* axis. The *x* axis indicates the congener numbers that are given in Appendix A. The dashed lines give the region within two standard deviations, and the solid lines depict the region within three standard deviations. The red bar represents the scaled concentration of congener 1234789-HpCDF.

**Figure 6 ijerph-16-04203-f006:**
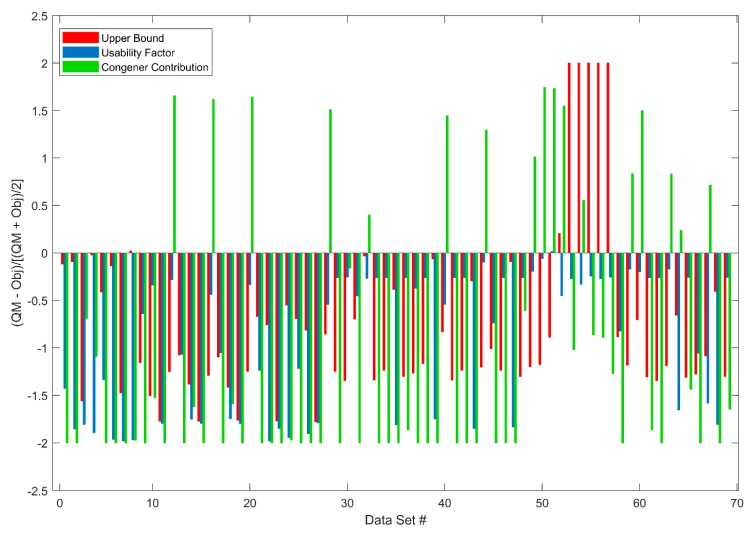
Difference between quality metrics and their objectives for upper bound (red bars), usability factor (blue bars), and congener contribution (green bars) are shown on the *y* axis for the 69 Target Data sets. The *x* axis gives the order of the 69 Target Data sets. For the same dataset, if all three bars are above zero, the dataset is not usable; otherwise, it is assessed as usable. #: Data Set.

**Figure 7 ijerph-16-04203-f007:**
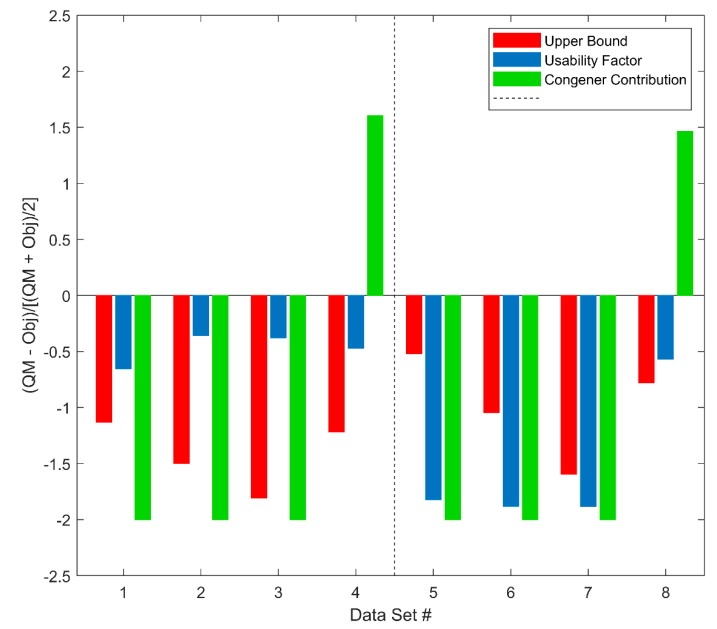
Difference between quality metrics and their objectives for upper bound (red bars), usability factor (blue bars), and congener contribution (green bars) are shown for four pairs of Target Data (left panel) and Duplicate Data (right panel). The *x* axis gives the order of the eight data sets. The eight datasets are from the whole milk tested on 29 June 2018. Datasets 1 and 5 are the detections of dioxins and furans; 2 and 6 are detections of mono PCBs; 3 and 7 are detections of marker PCBs, and 4 and 8 are detections of PBDEs. #: Data Set.

**Figure 8 ijerph-16-04203-f008:**
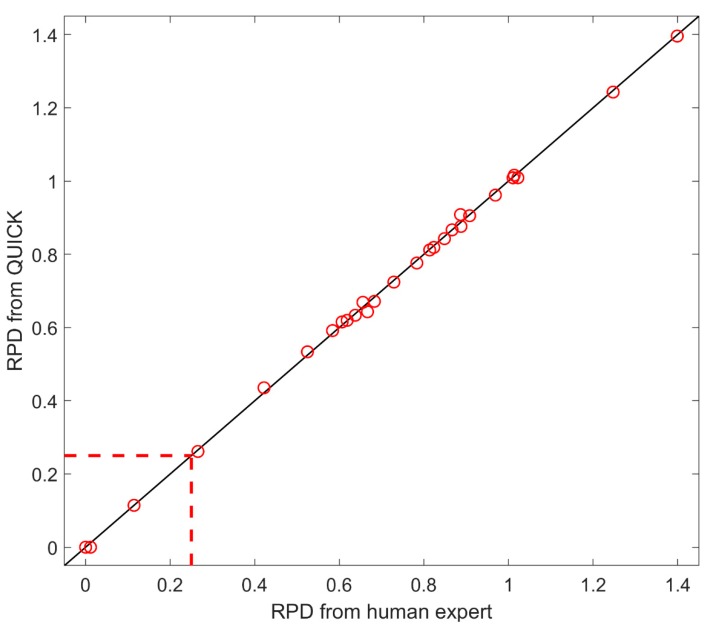
Comparison of relative percent difference (RPD) between human expert (*x* axis) and QUICK (*y* axis) for different congeners. Each circle represents a congener. If a circle is located on the solid diagonal line, the congener has the same RPD from QUICK and the human expert. The dashed lines represent the threshold at 0.25 and divide the plot into two regions. The bottom left region has 17 congeners that meet the criterion of RPD < 0.5, while the upper right region contains 25 congners that do not meet the criterion.

**Table 1 ijerph-16-04203-t001:** Objectives of sample upper bound.

Congener Type	TDS Sample	Chicken Egg Sample	Whole Milk Sample	Unit
Dioxin/Furan	0.1500	0.1800	0.0298	TEQ pg/g
mono PCB	0.0030	0.0037	0.0005	TEQ pg/g
marker PCB	500	437	40.68	pg/g
PBDE	150	150	150	pg/g

TDS: total diet study; PCB: polychlorinated biphenyls; PBDE: polybrominated diphenyl ethers; TEQ: toxic equivalency; pg: picogram; g: gram.

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
