# Peer review of "QUICK: Quality and Usability Investigation and Control Kit for Mass Spectrometric Data from Detection of Persistent Organic Pollutants"

_ijerph, 2019, doi:10.3390/ijerph16214203_

Round 1

Reviewer 1 Report

Guo et al., carried out a very interesting study on the development of the QUICK software kit. The designed tool is of much convenience and signficance since it can save much analysis time. This is a very novel research and can facilitate the analysis. The manuscript was  also well organized and well written, therefore, I strongly recommend it to be published in International Journal of Environmental Research and Public Health after minor revision.

1.  For line 15, * Correspondence:Correspondence: [email protected]; please delete one "Correspondence: "
2. Lin 3, is it possible to change 'in' to 'from'.  the data are generated from the detection of pops.

3. line 18, are -> cause

 4. Line 19, please delete 'in animals'. Since the mentioned properties are not only from animals., also from other environmental matrix.

5. Line 26, 'A diverse set of samples', please give the number of samples.

6. line 31, the key word is not necessary, since it appears in title, 'QUICK (Quality and Usability Investigation and Control Kit)'

7. For the method section, the type of samples were mentioned. Which type of samples should be given.

8. Please double check the whole manuscript for potential grammar errors and typos.

Author Response

We appreciate the reviewer for the constructive comments. We carefully read the comments and revised our manuscript accordingly. The reviewer’s comments dramatically improved our manuscript. Below we give the point-by-point responses to the reviewer’s comments. We hope our responses satisfy the reviewer.

   1. For line 15, * Correspondence:Correspondence: [email protected]; please delete one "Correspondence: "

[Author reply] Thanks for catching this. We deleted “Correspondence: ”.

Line 3, is it possible to change 'in' to 'from'. the data are generated from the detection of pops.

[Author reply] Thanks for the suggestion. We changed “in” to “from”.

line 18, are -> cause

[Author reply] Thanks for the suggestion. We made the change accordingly.

Line 19, please delete 'in animals'. Since the mentioned properties are not only from animals., also from other environmental matrix.

[Author reply] Thanks for the suggestion. We deleted “in animals”.

Line 26, 'A diverse set of samples', please give the number of samples.

[Author reply] Thanks for the comment. The number of samples (45) was added.

line 31, the key word is not necessary, since it appears in title, 'QUICK (Quality and Usability Investigation and Control Kit)'

[Author reply] Thanks for the suggestion. We deleted this key word.

For the method section, the type of samples were mentioned. Which type of samples should be given.

[Author reply] Thanks for the comment. In detection of persistent organic pollutants, different types of QC samples were used for different purposes. Four types of quality control samples (Daily Standard, Blank, Method Spike, Duplicate) are usually used to QC the experiment and assess data usability. Some or all of the samples are used in an experiment, depending on the instrument status.

Please double check the whole manuscript for potential grammar errors and typos.

[Author reply] Thanks for the suggestion. We have checked the whole manuscript and corrected the grammar errors.

Reviewer 2 Report

Line 152: just want to clarify that " mean + 2SD"is correct instead of "mean +/- 2SD"

Line 180: is there a reference for these EPA limits?

Line 207: Don't see a Table 2 anywhere

Line 246: add at end of sentence "is also available" current wording doesn't complete sentence/thought.

Line 257: any more information available about this standard sample? 

Line 260: reword, not clear that numbers on x-axis correspond to the congeners listed in Table S1

Line 301/302: Says dashed lines represent mean +/- 3SD, but 3 SD is marked with solid line on Figure 5

Figure 6:In the description of this data I think there could be more information as to what the cutoffs are (at what value on the y-axis is there a concern). A little more clarification here will really helps readers to better understand. 

Figure 7: specify what the samples are

Line 402: automation to automated

Author Response

We appreciate the reviewer for the constructive comments. We carefully read the comments and revised our manuscript accordingly. The reviewer’s comments dramatically improved our manuscript. Below we give the point-by-point responses to the reviewer’s comments. We hope our responses satisfy the reviewer.

Line 152: just want to clarify that " mean + 2SD"is correct instead of "mean +/- 2SD"

[Author reply] Thanks for the comment. Yes, it is “mean + 2SD”. Since Blank is an analyte free sample and it is used to make sure that the system is free of contaminants. A large concentration of a congener detected in the Blank sample (mean + 2SD) indicates low quality of sample extraction.  

Line 180: is there a reference for these EPA limits?

[Author reply] Yes, there is. The following reference (reference 12) is added: EPA Method 1613 Tetra- through Octa-Chlorinated Dioxins and Furans by Isotope Dilution HRGC/HRMS. US EPA Office of Water: Washington D.C., 1994.

Line 207: Don't see a Table 2 anywhere

[Author reply] Sorry! This is a typo. It should be Figure 2. It has been corrected in the manuscript.

Line 246: add at end of sentence "is also available" current wording doesn't complete sentence/thought.

[Author reply] Thanks for the suggestion. We reworded the sentence to the following. “The source code and Windows executable files are available upon request.”

Line 257: any more information available about this standard sample?

[Author reply] Thanks for the comment. We changed the sentence to the following sentence. “Here, the standard sample P180629-CS3A (tested on June 29th,2018 in FDA Dioxin lab) was taken as an example to show the performance of QUICK.”

Line 260: reword, not clear that numbers on x-axis correspond to the congeners listed in Table S1

[Author reply] Thanks for the comment. We changed the sentence to “The data quality metrics of the standard sample P180629-CS3A calculated by QUICK are shown in Figure 3 where the congener numbers on the x-axis are given in the first column of supplementary Table S1.”

Line 301/302: Says dashed lines represent mean +/- 3SD, but 3 SD is marked with solid line on Figure 5

[Author reply] Thanks for the comment. We changed “dash” to “solid”

Figure 6:In the description of this data I think there could be more information as to what the cutoffs are (at what value on the y-axis is there a concern). A little more clarification here will really helps readers to better understand.

[Author reply] Thanks for the comment. We added the following sentence to clarify this. “For the same dataset, if all three bars are above zero, the dataset is not usable, otherwise, it is assessed as usable.” 

Figure 7: specify what the samples are

[Author reply] Thanks for the comment. We added the following explanation in the manuscript. “The 8 datasets are from the whole milk tested on June 29th, 2018. Datasets 1 and 5 are the detections of dioxins and furans; 2 and 6 are detections of mono PCBs; 3 and 7 are detections of marker PCBs, and 4 and 8 are detections of PBDEs.”

    10.Line 402: automation to automated

[Author reply] Thanks for the comment. It has been corrected.